# Noble Metal Nanoparticles Incorporated Siliceous TUD-1 Mesoporous Nano-Catalyst for Low-Temperature Oxidation of Carbon Monoxide

**DOI:** 10.3390/nano10061067

**Published:** 2020-05-30

**Authors:** Badria M. Al-Shehri, Mohd Shkir, A. S. Khder, Ajeet Kaushik, Mohamed S. Hamdy

**Affiliations:** 1Catalysis Research Group (CRG), Department of Chemistry, College of Science, King Khalid University, P.O. Box 9004, Abha 61413, Saudi Arabia; balshehre@kku.edu.sa; 2Chemistry Department, College of Science, Umm Al-Qura University, Makkah 21421, Saudi Arabia; askhder@gmail.com; 3Advanced Functional Materials and Optoelectronics Laboratory (AFMOL), Department of Physics, College of Science, King Khalid University, Abha 61413, Saudi Arabia; shkirphysics@gmail.com; 4NanoBioTech Laboratory, Department of Natural Sciences, Division of Sciences, Art and Mathematics, Florida Polytechnic University, Lakeland, FL 33805, USA

**Keywords:** nano-catalyst, noble metals, TUD-1, mesoporous silica, CO-oxidation, air purification

## Abstract

This report, for the first time, demonstrated the low-temperature oxidation of carbon monoxide (CO) using nano-catalysts consisting of noble metal nanoparticles incorporated in TUD-1 mesoporous silica nano-structures synthesized via a one-pot surfactant-free sol–gel synthesis methodology. Herein, we investigated a nano-catalyst, represented as M-TUD-1 (M = Rh, Pd, Pt and Au), which was prepared using a constant Si/M ratio of 100. The outcome of the analytical studies confirmed the formation of a nano-catalyst ranging from 5 to 10 nm wherein noble metal nanoparticles were distributed uniformly onto the mesopores of TUD-1. The catalytic performance of M-TUD-1 catalysts was examined in the environmentally impacted CO oxidation reaction to CO_2_. The catalytic performance of Au-TUD-1 benchmarked other M-TUD-1 catalysts and a total conversion of CO was obtained at 303 K. The activity of the other nano-catalysts was obtained as Pt-TUD-1 > Pd-TUD-1 > Rh-TUD-1, with a total CO conversion at temperatures of 308, 328 and 348 K, respectively. The Au-TUD-1 exhibited a high stability and reusability as indicated by the observed high activity after ten continuous runs without any treatment. The outcomes of this research suggested that M-TUD-1 are promising nano-catalysts for the removal of the toxic CO gas and can also potentially be useful to protect the environment where a long-life time, cost-effectiveness and industrial scaling-up are the key approaches.

## 1. Introduction

The emission of carbon monoxide (CO) from several industrial applications and automobiles into atmosphere has been considered as one of the harmful environmental pollutions which indirectly affect global warming [1]. Due to continuous emission mainly via automobiles, in developed countries, the CO concentration has already reached an alarming level of about 65% [2]. Almost 90% of such emission is emitted during the cold step [3,4]. This means that the CO oxidation reaction can be a crucial step to minimize the emission of CO in the air and can therefore reduce the environmental pollutions. As one of the efficient solutions, several catalytic systems investigated to date for CO oxidation have been based on smart functional materials [5]. The observed significant outcomes have incited scientists to focus on design and develop novel nano-catalysts based on transition metals with half-filled d-orbital as the catalyst. This is due to their high activity on these nano-systems which are produced from their surface propensity to forming oxygen active species over the metal surface [6,7,8]. Considering this, noble metal-based nano-catalysts such as Au, Pt, Pd, Rh, Ru and Ag are being very effective to catalyze the CO oxidation reaction. However, the requirement of high temperature and pressure to achieve a better performance was a major concern in exploring them for wider applications [9,10]. Furthermore, the high price of noble metals also limited their utilization for CO oxidation applications if important commercial scale, such as for automobiles, is a point of focus. Thus, investigating noble metals (as dopants) supported that the transition metal-based nano-catalysis to be one of the best ways to overcome the challenges of high cost and tough operational parameters [11]. In addition, this new direction of developing nano-catalysts can lead to achieving a high activity in CO oxidation at a mild condition [12]. The quality of support toward CO oxidation was found to be dependent on various salient features such as porosity, productivity, selectivity and production cost along with reaction time, temperature and pressure [13,14]. Considering all of these aspects, developing noble metal-supported transition metals (metal oxides) such as [15,16] zeolite- [17] and clay-based [18] nano-catalysts will be of high significance to achieve CO oxidation in a controlled manner.

Reports have suggested that one of the effective supports recommended to accommodate noble metal nanoparticles with reference to catalytic activity is mesoporous silica. For example, silver (Ag) nanoparticles were doped in mesoporous silica materials of different morphology such as KCC-1, SBA-15, and MCM-41, to fabricate Ag/KCC-1. This nano-catalytic system exhibited the enhanced activity as noted in [19,20] as an indication of the influence of support on the catalytic activity of the Ag nanoparticles. In another study, the distribution degree of ruthenium (Ru) nanoparticles into different types (1D, 2D, 3D) of mesoporous silica, to fabricate Ru/SBA-15, Ru/KIT-6, Ru/MCM-41 and Ru/MCM-48 catalytic systems, was conducted to explore variation in CO oxidation [21]. In support of the above hypothesis, several researchers have also explored the effect of the active site available during the oxidation phenomena. For example, Ratnasamy et al. [22] incorporated gold (Au) and platinum (Pt) nanoparticles into MCM-41 mesoporous silica with different ratios for CO preferential oxidation, as the total CO conversion was at 80 °C. In addition, Ag, Pt and Rh were loaded into SBA-15 to investigate the activity and selectivity of the catalysts towards CO oxidation [23,24].

Considering the challenges and possible alternatives, this research, for the first time, demonstrated the influence of a 3D-type of mesoporous silica i.e., TUD-1 nanostructures doped with various noble metal nanoparticles (Au, Ru, Au, and Pt), on CO conversion to CO_2_ at a low temperature. These four nano-catalyst systems were synthesized via incorporating the selected noble metals into a TUD-1 framework using a one-step sol–gel technique. The developed nano-catalysts were chemically and physically characterized, the catalytic activity of the prepared catalysts was evaluated in CO oxidation reaction and the stability of the best performing catalyst was also investigated. 

## 2. Experiment

### 2.1. The Catalysts Synthesis

Four catalysts were synthesized by a one-pot surfactant-free sol–gel method which was reported earlier in [25], with a Si/M fixed ratio equal to 100 (M = Rh, Pd, Pt and Au). In typical procedure and based on the molar oxide ratio of 1SiO_2_:0.01MO_x_:0.5TEAOH:1TEA:15H_2_O, a mixture of triethanolamine (TEA, 97%, ACROS, Geel, Belgium) and noble metal solution (Rh(NO_3_)_2_·6H_2_O, Pd(NO_3_)_2_·6H_2_O, Pt(NO_3_)_2_·6H_2_O, or HAuCl_4_·XH_2_O, Sigma Aldrich, St. Louis, MO, USA) was slowly added under vigorous stirring of tetraethylorthosilicate (TEOS, +98%, ACROS, Geel, Belgium). After stirring for a few minutes, tetraethylammonium hydroxide (TEAOH, 35%, ACROS, Geel, Belgium) was added and the overall resultant mixture was kept under vigorous stirring for 2–24 h at ambient conditions until the gelation was achieved. The formed gel was dried overnight at 371 K, and then hydrothermally treated in stainless autoclaves for four hours at 451 K. Finally, the produced powders were calcinated for ten hours at 873 K with a programed heating of 1 K/min. The final solid powders were grinded, sieved and stored in clean desiccator.

### 2.2. The Catalysts Characterizations 

The four factionalized Rh-TUD-1, Pd-TUD-1, Pt-TUD-1 and Au-TUD-1 nano-systems were characterized using a set of elemental and quantitative tools. The amount of metals was measured by a inductively coupled plasma (ICP-OES) analysis of Thermo scientific (Waltham, MA, USA), ICAP 7000 series, part No: 1340910, Qtegra Software (1.3.882.20), (Waltham, MA, USA). The degree of crystallinity was determined using powder X-ray diffraction (XRD) patterns which were recorded on a Schimadzu 6000 DX diffractometer (Shimadzu Corporation, Kyoto, Japan) equipped with a graphite monochromator using Cu Kα (*λ* = 0.1541 nm). The Nitrogen adsorption–desorption isotherms were recorded on a QuantaChrome Autosorb-6B at 77 K (Quantachrome Instruments, Boynton Beach, FL, USA). The pore size distribution was calculated from the adsorption branch using the Barret–Joyner–Halenda (BJH) model. The Brunauer–Emmett–Teller (BET) method was used to calculate the surface area of the samples, whereas the mesopore volume and external surface area were calculated using the *τ*- plot method. Scanning electron microscopy (SEM, Jeol Model 6360 LVSEM, JEOL Inc., Peabody, MA, USA) was used to investigate the morphological structure, and the micrographs were obtained after coating the samples with gold to avoid charging affects and to enhance contrast. Moreover, the SEM was equipped with an energy dispersive Xray (EDX, Jeol Model 6360 LVSEM, JEOL Inc., Peabody, MA, USA), which was used for qualitative and semi-quantitative elemental analysis. High-resolution transmission electron microscopy (HR-TEM) was carried out on a Philips CM30UT electron microscope (Philips Electronics, Eindhoven, The Netherlands) with a field emission gun as the source of electrons, and was operated at 300 kV. Finally, the chemical state of the metals and the surface composition of materials were analyzed by a X-ray photoelectron spectroscopy (XPS) PHI 5400 ESCA (ULVAC-PHI, Inc., Kanagawa, Japan) equipped with a dual Mg/Al anode X-ray source, a hemispherical capacitor analyzer and a 5 keV ion gun. All measurements were carried out at room temperature.

### 2.3. Catalytic Activity Investigation

The catalytic oxidation of CO was carried out in a fixed bed reactor. In this system, there were three units, the first was the mixing and feed unit which was responsible for the gases mixing, the second unit was the reactor unit which contained the fixed-bed quartz tube, while the third unit was the analysis unit which contained the gas chromatograph (GC) equipped with a thermal conductivity detector (TCD). According to the classical procedure of the experiment, 50 mg of the prepared catalyst, with a particle size between 180 and 315 μm to minimize the mass transfer limitation, was loaded into the middle of a quartz tube without dilution and with a space velocity (GHSV) of 36,000 h^−1^ at 3 mbar. A pre-treating process for the catalyst was necessary to clean the surface and to activate the catalyst, which was performed by heating the reactor to 673 K for 1 h and then cool it back to the initial temperature. Then, a mixture of 1% CO and O_2_ (both balanced with N_2_ gas) was feeding into the reactor with a pressure of 4.5 mbar. The temperature program was adjusted to start heating from room temperature to 673 K with a heating ramp of 1 K/min. Finally, the concentrations of the gas mixture i.e., CO, CO_2_, O_2_ and N_2_, were analyzed online by gas chromatograph (Chrompack CP9001, Varian, Middelburg, The Netherlands) equipped with a TCD detector by using the Poraplot Q column.

## 3. Results and Discussion

### 3.1. Compositional Analysis 

The ICP elemental analyses were performed to quantitatively determine the amount of noble metal in each sample. Rh-TUD-1 showed the maximum ions incorporation (98.8%), while Au-TUD-1 exhibited the lowest ions incorporation with a percentage of 93.4%. The obtained results are listed in Table 1 and it showed that more than 93% of all the added noble metal ions in the synthesis mixture were found in the final solid product.

The textural properties of the TUD-1 and M-TUD-1 catalysts were determined using a N_2_ physisorption analysis approach and the outcomes are summarized in Table 1. All the prepared samples exhibited a high specific surface area of more than >620 m^2^/g. The Pt-TUD-1 exhibited the highest surface area (710 m^2^/g), while the Pd-TUD-1 showed the minimal surface area (622.3 m^2^/g). The reason for the difference of surface area could be related to the size and distribution of the nanoparticles present inside the pores of TUD-1. Moreover, the pore volumes of all four M-TUD-1 samples were less than that of the neat TUD-1 sample, which gave an indication of the presence of noble metal nanoparticles inside the mesopores of TUD-1. On the other hand, the pore size of the entire M-TUD-1 samples were larger than that of the TUD-1 as an indication of the expansion of the pores during the synthesis to accommodate the formed noble metal nanoparticles.

Moreover, the N_2_ sorption isotherms of the prepared catalysts were compared with reference to TUD-1 as shown in Figure 1A. The M-TUD-1 samples displayed a typical IV isotherm with H_2_-hysteresis loop, which was very similar to that of the TUD-1 isotherm which suggested that the prepared nano-catalysts belonged to the mesoporous category as per the approved IUPAC concept [26]. The corresponding pore size distribution (Figure 1B) showed a narrow and non-uniformed pore size as well as shape distribution signals in the range of 4.6 to 7.5 nm. Moreover, it is worth noting that the pore diameter of these catalysts belonged to the mesopore category (2–50 nm) [27].

To investigate the crystallinity of all four M-TUD-1 nano-catalysts, the XRD patterns were recorded and compared (as presented in Figure 2A) with the TUD-1 material as a reference. The TUD-1 and the M-TUD-1 catalysts exhibited a broad diffraction peak ranging from 20° to 30° which indicated the formation of amorphous matrices of silicates according to previous reports [28]. This indicated that the framework of the TUD-1 did not significantly changed after the incorporation of the noble metal nanoparticles. In M-TUD-1, new additional sharp peaks appeared, which typically referred to the face-centered cubic (fcc) crystalline skeleton of the (111), (200), and (220) planes [29,30]. In these diffraction patterns, the (111) plane referred to the strongest diffraction peaks at 41.07°, 40.05°, 40.0° and 38.33° for Rh, Pd, Pt and Au, respectively [31]. However, the second diffraction peaks at 47.95°, 44.58°, 46.03° and 44.8° were assigned to the (200) plane of noble metals with the same order. Finally, weak diffraction peaks around 70.14°, 68.47°, 67.54° and 65.01° were identified as the (220) plane of M = Rh, Pd, Pt and Au, respectively [32,33].

Figure 2B shows the SEM micrographs of the four M-TUD-1 nano-catalysts. All the nano-catalyst particles exhibited the irregular shape, which is a characteristic of amorphous TUD-1 mesoporous material. Furthermore, no bulky or separating crystals were observed which can be indicative of the total incorporation of noble metals into the TUD-1 framework. The size and the distribution of the noble metal nanoparticles were determined through the HR-TEM technique as illustrated in Figure 2C. The four micrographs exhibited a worm-like porous network of the TUD-1, and the well distributed noble metal nanoparticles are clearly visible. The average size of nanocrystals was almost in the range of 5 to 10 nm. More importantly, the obtained micrographs did not show any nanoparticle aggregation or bulky phase as an indication of the high validity of the synthesis procedure to prepare such types of catalytic structures.

Moreover, the elemental mapping analysis of the four M-TUD-1 nano-catalysts was explored using EDS as presented in Figure 3. A uniform distribution of the incorporated noble metals (Rh, Pd, Pt and Au) was observed in all the nano-catalyst samples. Moreover, the EDX outcomes confirmed the presence of Si, O, in addition to the incorporated noble metal (Rh, Pd, Pt or Au) only without strange ions which was evidence of the high purity of the prepared samples.

The chemical states of the surface elements in addition to the chemical compositions of the catalysts were investigated by using the XPS technique. The obtained results are presented in Figure 4 and the corresponding binding energies are listed in Table 2. For all the samples, the two peaks which were located at values of about 103.85 eV and 533.01 eV were attributed to Si 2p and O 1s, respectively, according to the previous reports [34,35]. The XPS profile of Rh-TUD-1 showed the binding energies of Rh 3d_5/2_ and Rh 3d_3/2_ around 307.6 eV and 311.8 eV, respectively, which could be assigned to the presence of Rh^0^ as a metallic nanoparticle on the mesoporous surface of the Rh-TUD-1 catalyst [36,37]. Additionally, in the spectrum of Pd-TUD-1, two peaks centered at 340 eV and 335.7 eV which could be assigned to Pd 3d_5/2_ and Pd 3d_3/2_ in the Pd^0^ state [38]. Furthermore, in the XPS spectrum of Pt-TUD-1, two main peaks located around 71.76 eV and 74.97 eV of Pt 4f_7/2_ and Pt 4f_5/2_ were clearly visible, which could be assigned to zero-valent platinum [39]. Finally, as can be seen in the XPS spectrum of Au-TUD-1, it can be observed that the peaks of Au 4f_7/2_ and Au 4f_5/2_ were located at 84.42 eV and 88.04 eV, which corresponds to Au^0^ state as reported previously [40]. 

### 3.2. Catalytic Performance toward CO Oxidation

The results of the CO oxidation reactions which were catalyzed by the different noble metals incorporated the TUD-1 mesoporous material as presented in Figure 5A. The obtained results showed that Au-TUD-1 exhibited the highest activity, in which a 100% conversion of CO was obtained at 303 K. On the other hand, Pt-TUD-1 exhibited a 100% conversion within 318 K, and Pd-TUD-1 also converted 100% of CO at 328 K, while Rh-TUD-1 converted 100% of CO at 348 K. The activity could be set in the order of Au > Pt > Pd > Rh, which was the same trend obtained by Santos et al. [41]. Therefore, the best performing catalyst, Au-TUD-1, was chosen for further investigations.

The reusability of the Au-TUD-1 sample was evaluated via activating ten consecutive reactions without treatment and the observed results are presented in Figure 5B. This nano-catalyst sample exhibited a high reusability property and maintained almost the same activity in the consecutive reactions, and 100% conversion of CO was obtained in the range of 303–305 K in all of the reactions. This result did not only show the high efficiency of Au-TUD-1, but also the good possibility of using it in many consecutive reactions without treatment.

In order to investigate the stability of the Au-TUD-1 sample, an additional XPS study was conducted for the sample which was used in the consecutive reactions. The obtained XPS is presented in Figure 5C. Two peaks were obtained at binding energies of 84.42 eV and 88.04 eV which assigned the Au 4f_7/2_ and Au 4f_5/2_, which could be indicative of the presence of Au^0^ nanoparticles in the silica matrix. By comparing the fresh XPS results (presented in Figure 4) and the XPS for the used sample, neither the positions nor the intensity of the peaks were changed which was an indication for the high stability of the oxidation state of gold nanoparticles. 

### 3.3. Proposed Mechanism

The mechanism of CO oxidation using molecular oxygen over noble metals involves several steps supported by the literature [42,43,44]. In general, M-TUD1-assisted CO oxidation is a three-step procedure (as illustrated in Figure 6A), as follows: (i) CO adsorption, (ii) O_2_ activation and (iii) CO oxidation. In brief, the first step included the direct and immediate adsorption of the CO gas molecules on the M^0^ active sites [42,43]. This was a chemisorption process which involved a strong chemical bond formed between the metal surface and the CO molecules. The second step was the activation of the O_2_ molecules, which took place through the formation of oxygen vacancies in the rich surface of the nano-catalyst, which accelerates the formation of the metal-O_2_ molecule. The key factor at this step was the requirement energy to dissociate the adsorbed oxygen molecules. Finally, as the third step, the CO_2_ formation took place as oxygen migration adsorbed by the CO molecule on the dual- or active-site of the catalyst (surface oxygen vacancies and nanoparticles of noble metals). 

### 3.4. The Differences in Activity

Based on the outcomes of the analytical studies, it was observed that there were significant differences in the efficiency of the four investigated noble metals toward CO oxidation. Although the largest surface area was for Pt-TUD-1 followed by Rh-TUD-1 and then Au-TUD-1, and finally, Pd-TUD-1, it was Au-TUD-1 that exhibited the highest activity. So, the surface area was not the only key factor for explaining such activity difference as reported in many previous articles [45,46,47]. On the other hand, the obtained activity difference would be explained by the distribution degree of the metal nanoparticles and their average size, but all the prepared catalysts exhibited almost the same trend in particle size and the same well distributed nanoparticles. Therefore, the metal distribution and the particle size were not the critical points in explaining the activity difference. 

Hence, the obtained activity difference must have been related to the metals themselves and their chemical properties. The key factor was most likely the activation energy of each metal as a function of oxygen chemisorption dissociation where the activation and migration facilities of oxygen species were conductive to the formation of abundant surface oxygen vacancies and played a significant role in the CO oxidation reaction at a lower temperature. Considering the dissociative adsorption of oxygen which is illustrated in Figure 6B, the specific activities at 175 °C of the CO reaction for the different prepared catalysts were in agreement with the obtained activity trend, where the Au-TUD-1 nanoparticles had the lowest value of activation energy required for oxygen gas dissociation with the highest TOF, followed by Pt-TUD-1, and the highest value of activation energy being in Rh-TUD-1. The obtained activity trend and the possible explanation were in good agreement with the research outcomes of Santos et al. [43]. 

### 3.5. Comparison with Other Supports

As observed, amongst several noble metals, Au nanoparticles were very active in the CO oxidation [45,46,47,48,49,50,51,52,53,54,55]. Several of evidenced noble metals were explored to support the claim, with reference to Au nanoparticles, that some of them were inert and did not participate in one way or another in the chemical reaction, e.g., silica [54,55], while some participated in the reaction such as Al_2_O_3_ in [51]. A comparison between the different materials which were used to support the Au nanoparticles in the CO oxidation reaction are listed in Table 3.

Based on the results listed in Table 3, the Au-TUD-1 exhibited high activity, not only more than other mesoporous silica materials such as MCM-41 and SBA-15, but also more than the other supporting materials such as CeO_2_, CuO and MnO_2_. The TUD-1 mesoporous material, with its uniform pore system and interconnectivity pores, offers excellent accessibility for reactants and products to reach or leave the Au^0^ active sites which makes the reaction proceed faster than for other mesoporous materials. Few studies were also reported to show the advantages of using TUD-1 as a support for different active sites, such as copper in phenol hydroxylation [56], vanadium in the epoxidation of trans-stilbene [57] and indium in Baeyer–Villiger oxidation [58]. Therefore, Au-TUD-1 was considered a promising catalyst for CO oxidation, and further developments were carried out to reach the conversion temperature below the room temperature and to compare it with other Au catalysts [59,60], which will be reported shortly.

## 4. Conclusions and Viewpoint

In summary, the design and development of a class of nano-catalysts including noble metal nanoparticles supporting mesoporous silica nano-catalysts using an easy and scalable method was demonstrated in this research. The catalytic performance of these nano-catalysts was evaluated using CO oxidation at a low temperature, as a model system. We observed that the M^0^ nanoparticles of (Rh, Pd, Pt or Au) with an average size of 5–10 nm were successfully incorporated into the siliceous TUD-1 mesoporous material by a one-step sol–gel surfactant-free procedure. The outcomes of analytical studies confirmed the formation of noble metal nanoparticles with the same loading (around 1%) and a high distribution into 3D-silica framework. Based on the obtained catalytic activities toward CO oxidation, the Au-TUD-1 catalyst exhibited the best catalytic performance for the oxidation of CO at room temperature followed by Pt-TUD-1 and then Pd-TUD-1, whereas Rh-TUD-1 exhibited the lowest activity at a higher temperature. The observed reusability of Au-TUD-1 projected this nano-catalyst’s useful for ten consecutive catalytic reactions without compromising its activity.

The herein developed nano-catalysts based on metallic nanoparticles are cost-effective and exhibit fascinating catalytic activity. These salient features project these nano-catalysts as desired materials with promising applications in air purification, respiratory/escape masks for removing gases, refuge chambers, devices for self-rescue breathing and several others to protect from harmful gases present in the environment. Such nano-catalysts are also suitable to design and develop filters or cartridges for the trapping of harmful particles of chemicals, substances and gases in the atmosphere where the living breath to live, work, or survive.

## Figures and Tables

**Figure 1 nanomaterials-10-01067-f001:**
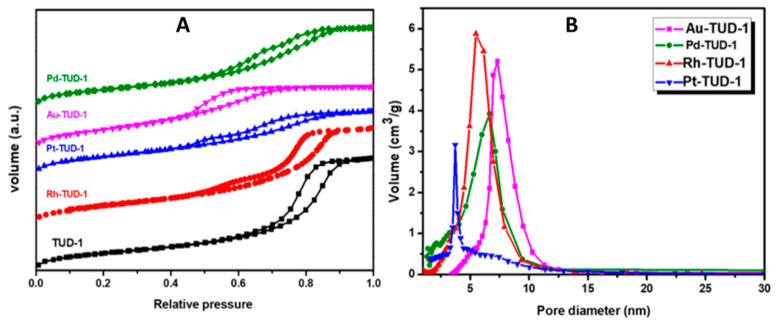
(**A**) N_2_ sorption isotherms of the four investigated M-TUD-1 catalysts as obtained from a QuantaChrome Autosorb-6B at 77 K. (**B**) The corresponding pore size distribution as calculated from the adsorption branch using the Barret–Joyner–Halenda (BJH) model.

**Figure 2 nanomaterials-10-01067-f002:**
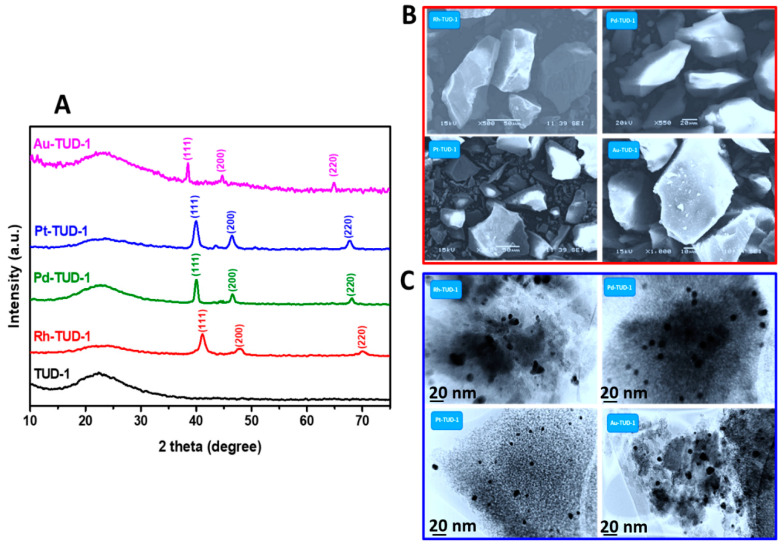
(**A**) XRD patterns of the four prepared M-TUD-1 (M = Au, Pt, Pd and Ru) samples compared with that of the TUD-1, (**B**) the SEM micrographs and (**C**) the high-resolution transmission electron microscopy (HR-TEM) micrographs of the M-TUD-1 (Rh, Pd, Pt and Au) samples.

**Figure 3 nanomaterials-10-01067-f003:**
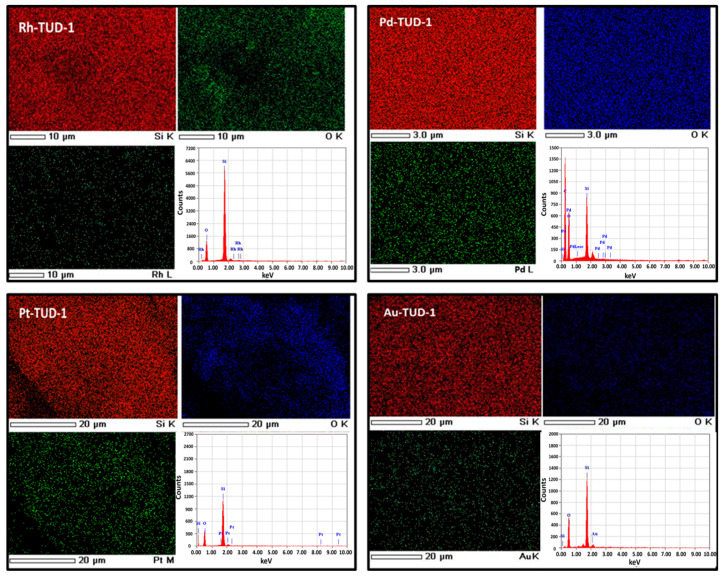
The elemental mapping and the energy dispersive Xray (EDX) spectroscopic analysis of the M-TUD-1 (Rh, Pd, Pt and Au) samples.

**Figure 4 nanomaterials-10-01067-f004:**
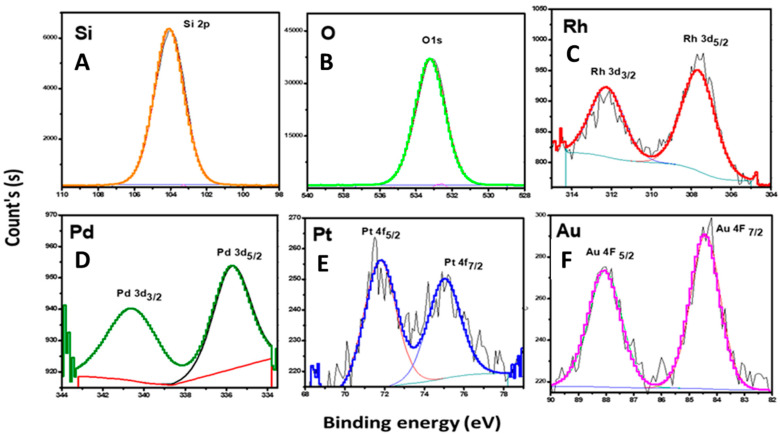
X-ray photoelectron spectroscopy (XPS) profiles of Si 2p (**A**), O 1s (**B**), Rh 3d (**C**), Pd 3d (**D**) Pt 4f (**E**) and of Au 4f (**F**) over the M-TUD-1 (where M = Rh, Pd, Pt, Au) catalysts.

**Figure 5 nanomaterials-10-01067-f005:**
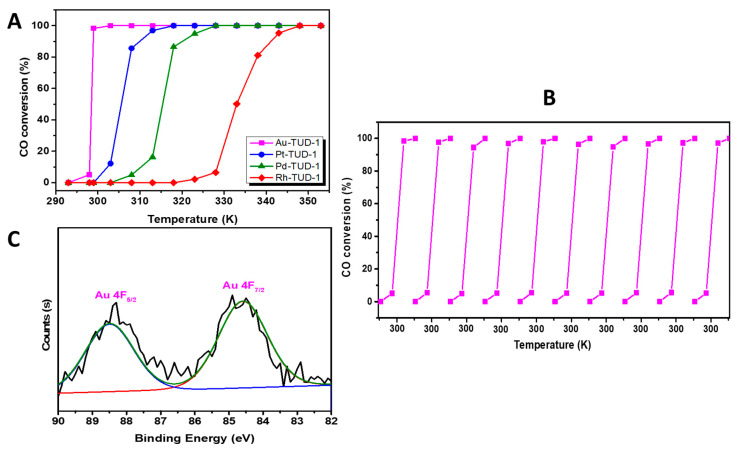
(**A**) The conversion of carbon monoxide (CO) in the presence of O_2_ as a function of the applied temperature over the four investigated M-TUD-1 catalysts. (**B**) The performance of the Au-TUD-1 catalyst in 10 consecutive runs represented in the CO conversion % as a function of the applied temperature and (**C**) the XPS of the Au-TUD-1 catalyst sample after ten consecutive runs.

**Figure 6 nanomaterials-10-01067-f006:**
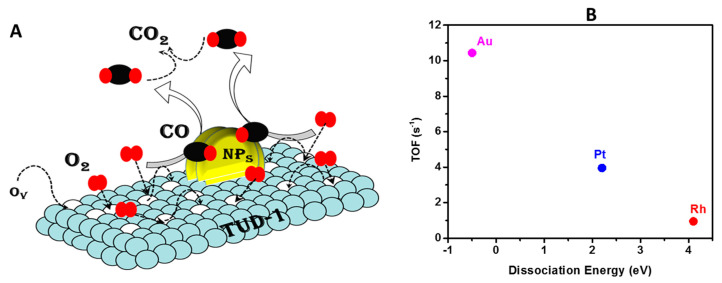
(**A**) The proposed mechanism for CO oxidation over M-TUD-1 and (**B**) the obtained turn over frequency (TOF) as a function of the dissociation energy of the oxygen molecules over the applied noble metal nanoparticles.

**Table 1 nanomaterials-10-01067-t001:** The elemental analysis and the texture properties of the prepared M-TUD-1 samples as obtained from the ICP and N_2_ sorption measurements, respectively.

Sample	Si/M Ratio		Texture Properties		
	Synthesis Mixture	Final Product	Surface Area (m^2^/g)	Pore Volume (cm^3^/g)	Pore Size (nm)
TUD-1	∞	∞	655.2	1.710	4.5
Pt-TUD-1	100	101.41	710.6	1.165	4.6
Rh-TUD-1	100	101.21	685.9	0.804	5.9
Au-TUD-1	100	107.06	644.1	0.970	7.5
Pd-TUD-1	100	103.62	622.3	1.291	6.8

**Table 2 nanomaterials-10-01067-t002:** The binding energy of the Si 2p, O 1s, Rh 3d, Pd 3d, Pt 4f and the Au 4f photoelectrons of the catalysts (experimental error = ±0.5 eV).

Peaks of XPS Spectrum	Binding Energy (eV)			
	Rh-TUD-1	Pd-TUD-1	Pt-TUD-1	Au-TUD-1
Si 2p	102.9	102.73	102.8	102.76
O 1s	531.9	531.93	531.97	530.68
Rh 3d	5/2 = 307.6 3/2 = 311.8	-	-	-
Pd 3d	-	5/2 = 335.7 3/2 = 340	-	-
Pt 4f	-	-	5/2 = 71.76 7/2 = 74.97	-
Au 4f	-	-	-	5/2 = 84.42 7/2 = 88.04

**Table 3 nanomaterials-10-01067-t003:** A comparison between the different materials which supported the Au nanoparticles in the CO oxidation and the obtained activity.

Ref.	Au Support	Au Loading %	Au Nanosize (nm)	Temperature (K)
Current work	TUD-1	0.94	5–10 nm	303 K
[45]	α-FeOOH	0.92	30 nm	507 K
[46]	mesoporous Fe_2_O_3_	7.9	3–10 nm	523 K
[47]	MnO_2_	1.0	1–3 nm	321 K
[48]	CeO_2_/Al_2_O_3_	1.6	2–3 nm	322 K
[49]	CuO	1.0	4–8 nm	353 K
[50]	CeO_2_	1.0	5–10 nm	393 K
[51]	Ni/Al_2_O_3_	1.0	2.4–3.5 nm	293 K
[52]	FeO_x_/TiO_2_	1.0	5 nm	373 K
[53]	Mn_3_O_4_	4.0	30–40 nm	371 K
[54]	SBA-15	4.8	10–50 nm	433 K
[55]	MCM-41	4	5.1–6.9 nm	454 K

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
