# Peer review of "Noble Metal Nanoparticles Incorporated Siliceous TUD-1 Mesoporous Nano-Catalyst for Low-Temperature Oxidation of Carbon Monoxide"

_nanomaterials, 2020, doi:10.3390/nano10061067_

Round 1
Reviewer 1 Report
The manuscript reports on the production of mesoporous nano-catalysts by incorporating through a solgel strategy noble metal nanoparticles into TUD-1 silica. The suitability of the obtained catalysts for the CO conversion to CO2 was also assessed by breakthrough experiments. The manuscript is well arranged scientifically and contains much new information, but an improvement of some sections should be provided. The paper contains useful information in the field and for these reasons, I consider the paper suitable for publication, but before this, some minor revisions should be done. I suggest to improve the entire manuscript taking into account the following indications:
Entire manuscript:
- The work plan is correct and the manuscript is easy to follow, but, anyway, I suggest a linguistic revision and a proper check for some grammar errors.
Experimental section:
- Line 92: I suggest to change the word morphology with crystallinity
- Information about the work out of XPS data should be reported.
- The authors reported in section 3.1 ICP-MS data, but they do not report related experimental details in the experimental section.
- Did the authors perform leaching test to probe the loss of the supported noble metal particles?
Results and Discussion section:
- Lines 139-145: I suggest to refer to IUPAC reference articles for the classification of adsorption isotherms and for the classification of pores into micro, meso and macropores.
- Section 4. In my opinion a separate Discussion section is not necessary, a combined section of Results and Discussion is better.
- Lines 242 and 265: please check the proposed sentences
- Which are the possible poisoning agents for the proposed catalysts? Did the authors test this aspect?
- Why the authors tested only one CO inlet concentration?
- Section 4.1: the description of the proposed mechanism is quite confused, please improve it.
- Section 4.3: I am not convinced at all that a comparison as the one proposed by the table 3 is enough to say that TUD-1 is a better support for noble metal nanoparticles than other metal oxides. The reported materials are similar to that proposed by the manuscript under review, but the catalytic tests have been performed in different ways so to me the comparison is not homogenous.
- Line 281: among the materials listed in Table 3, which is the one containing a MOF as supporting agent?
Author Response
Response to reviewers #
Manuscript ID: nanomaterials-800552
Manuscript Title: Noble metals nanoparticles incorporated siliceous TUD-1 mesoporous nano-catalyst for low-temperature oxidation of carbon monoxide
Authors response
The authors are thankful to the editor and reviewers for their constructive and motivating and comments on our manuscript “nanomaterials-800552”.
Please be noted that the all the comments of editor and reviewer have been addressed in the revised manuscript point-by-point, as described below
Point-by-point response to Reviewer’s comment
Reviewer #1
The manuscript reports on the production of mesoporous nano-catalysts by incorporating through a solgel strategy noble metal nanoparticles into TUD-1 silica. The suitability of the obtained catalysts for the CO conversion to CO2 was also assessed by breakthrough experiments. The manuscript is well arranged scientifically and contains much new information, but an improvement of some sections should be provided. The paper contains useful information in the field and for these reasons, I consider the paper suitable for publication, but before this, some minor revisions should be done. I suggest to improve the entire manuscript taking into account the following indications:
Entire manuscript:
The work plan is correct, and the manuscript is easy to follow, but, anyway, I suggest a linguistic revision and a proper check for some grammar errors.
Experimental section:
Comment 1- Line 92: I suggest to change the word morphology with crystallinity
Response 1: The word morphology is changed to crystallinity
Comment 2- Information about the work out of XPS data should be reported.
Response 2: The information of the XPS instrument is added to the section of 2.2.
Comment 3- The authors reported in section 3.1 ICP-MS data, but they do not report related experimental details in the experimental section.
Response 3: The ICP analysis details is added to the experimental section.
Comment 4- Did the authors perform leaching test to probe the loss of the supported noble metal particles?
Response 4: The authors did not perform leaching test.
Results and Discussion section:
Comment 5- Lines 139-145: I suggest to refer to IUPAC reference articles for the classification of adsorption isotherms and for the classification of pores into micro, meso and macropores.
Response 5: Reference 26 is changed to another appropriate reference.
Comment 6- Section 4. In my opinion a separate Discussion section is not necessary, a combined section of Results and Discussion is better.
Response 6: The results and discussion sections are combined in one section.
Comment 7- Lines 242 and 265: please check the proposed sentences.
Response: Corrections have been made in revised manuscript.
R7: We have checked and modified the above-mentioned sentences
Response: Corrections have been made in revised manuscript.
Comment 8- Which are the possible poisoning agents for the proposed catalysts? Did the authors test this aspect?
Response 8: The authors did not investigate the poisoning agents because the most active catalyst; Au-TUD-1, was reused efficiently for several times without activity loss. However this will be considered in the future study especially in the presence of sulfur/sulfur compounds.
Comment 9- Why the authors tested only one CO inlet concentration?
Response 9: The aim of the current study was to investigate the activity difference between the supported Noble metals. The coming study will involve the optimization of different catalyst preparation factors as well as the operational parameters.
Comment 10- Section 4.1: the description of the proposed mechanism is quite confused, please improve it.
Response 10: Description of proposed mechanism has been modified which is now changes to section 3.3.
Comment 11- Section 4.3: I am not convinced at all that a comparison as the one proposed by the table 3 is enough to say that TUD-1 is a better support for noble metal nanoparticles than other metal oxides. The reported materials are similar to that proposed by the manuscript under review, but the catalytic tests have been performed in different ways so to me the comparison is not homogenous.
Response 11: The authors agree with the reviewer about his/her opinion that the reactions conditions are different indeed. But the main purpose of the comparison was to present the different supports which were used to accommodate the Au nanoparticles. The aim was not to show that TUD-1 is the best support because the authors presented a more active catalytic system Ref [51].
Comment 12- Line 281: among the materials listed in Table 3, which is the one containing a MOF as supporting agent?
Response 12: The authors apologized for this mistake, MOF was removed from the text.
Reviewer #2
Comment The manuscript reports fabrication of noble metal-decorated silica and their use for CO oxidation. The results show that Au is the most efficient catalyst for the photocatalytic reaction with 100% conversion at 303 K. The manuscript can be accepted for publishing after the following points are addressed:
Response: The authors thank the reviewer for his/her positive opinion about the manuscript
Comment 1- Why is there a difference in the specific surface area among noble metal samples?
Response 1: The difference can be related to the distribution and the size of the nanoparticles incorporated in the silica matrix. The reason is added to the text.
Comment 2- What is the scale bar in TEM images?
Response 2: The scale bar has been mentioned on TEM images
Comment 3- How did the authors obtain dissociation energy?
Response 3: The dissociation energy was obtained from the Ref [43].
Comment 4- How did the authors change the temperature of the reactor?
Response 4: The temperature program was from room temperature to 673K with a heating ramp of 1K/min. The reaction temperature is added to experimental section.
Comment 5- The axis in EDS spectrum is not legible.
Response 5: Axis of EDX spectrums are redrawn to make more visible
Comment 6- Page 2 line 70, is this M-TUD-1 “doped” or deposition?
Response 6: The authors think that doped is more appropriate word.
Comment 7- Typo: not necessary to capitalize “noble metal”, table 3 “current work”.
Response 7: The word noble is corrected through the text. Current work is corrected in table 3.
Comment 8- Other works on CO oxidation at low temperature using Au: ACS Catal. 2019, 9, 9, 8364-8372; Materials Research Bulletin 2019, 115, 247-256.
Response 8: The two references are added to the texted as Ref [59] and [60].

Reviewer 2 Report
The manuscript reports fabrication of noble metal-decorated silica and their use for CO oxidation. The results show that Au is the most efficient catalyst for the photocatalytic reaction with 100% conversion at 303 K. The manuscript can be accepted for publishing after the following points are addressed:
- Why is there a difference in the specific surface area among noble metal samples?
- What is the scale bar in TEM images?
- How did the authors obtain dissociation energy?
- How did the authors change the temperature of the reactor?
- The axis in EDS spectrum is not legible.
- Page 2 line 70, is this M-TUD-1 “doped” or deposition?
- Typo: not necessary to capitalize “noble metal”, table 3 “current work”.
- Other works on CO oxidation at low temperature using Au: ACS Catal. 2019, 9, 9, 8364-8372; Materials Research Bulletin 2019, 115, 247-256.
Author Response

(The authors gave the same response as above.)
